# Green Methane as a Future Fuel for Light-Duty Vehicles

Jaewon Byun [1] and Jeehoon Han [2,*]

1   Petrochemical Material Engineering Department, Chonnam National University,
    Yeosu 59631, Republic of Korea
2   Department of Chemical Engineering, Pohang University of Science and Technology,
    Pohang 37673, Republic of Korea
*   Correspondence: jhhan@postech.ac.kr; Tel.: +82-54-279-2274

**Abstract:** Food waste (FW) has traditionally been disposed by incineration or landfilling; however, it can be converted to green methane (GM) via anaerobic digestion, and GM can be used as fuel for light-duty natural gas vehicles (LDNGVs). A lifecycle assessment (LCA) of FW-based GM production and LDNGV operation in China, a new scenario, was performed. The LCA results were compared with those for the conventional FW treatment, where a "well-to-wheel" system boundary including FW collection, GM production from FW, and vehicle manufacturing, operation, and disposal was defined. The LCA results showed that the global warming impacts of the new FW scenario are 44.3% lower than those of the conventional option. The fine particulate matter formation impact of the new FW scenario was dominated by the displacement effect of electricity supply to anaerobic digestion, followed by $CO_2$ adsorption by the primary source. The sensitivity analysis showed that hydroelectric power as the best primary source for electricity supply could substantially reduce both global warming and FRS in the new scenario. In the short term, the proposed FW scenario could be a feasible option for achieving sustainable society by minimizing environmental impacts of FW treatment.

**Keywords:** life cycle assessment; methane; anaerobic digestion; food waste; NGV

## 1. Introduction

Food waste (FW) is a biodegradable waste that consists mainly of carbohydrates, proteins, and lipids [1–3]. Its composition varies with respect to food type (e.g., meat, vegetables, rice, or milk) and food source (e.g., food processing industries or households) [4]. The amount of FW generated per year has been steadily increasing, which is expected to reach 3.62 billion tons in 2030—a substantial increase from the 1.95 billion tons generated in 2011 [4–7]. China is the one of the top FW-generating countries in the world (1.17 billion tons; 32.2% of global total) [4]. China's FW consists mainly of rice and vegetables and is rich in carbohydrates [4]. Conventionally, this FW is landfilled or incinerated (greater than 90%) [8], which can lead to severe health and environmental problems such as greenhouse gas (GHG) emission, particulate matter generation, etc. [9,10]. Additionally, the conventional FW treatment reduces the economic value of the substrate because it hinders the recovery of a valuable substrate to be recycled. The FW contains carbohydrate, protein, and lipid, and they can be converted to valuable chemicals or fuels by several conversion technologies. Anaerobic digestion (AD), which converts FW to biogas, is potentially an appropriate method to effectively manage FW and strengthen a nation's energy security in that heat and power can be produced from biogas [11,12]. The production rate of biogas based on the current AD technology is 0.083–0.256 $m^3$ per kg of dry FW and day, and the China has the largest bioenergy potential (33.9–104.5 million $m^3$ of biogas per year) from the AD of FW. Previous studies [2,3,13] analyzed economic and environmental feasibility of biogas production from solid wastes, and presented the benefit of the biogas production by AD for waste treatment.

Biogas, which is a methane ($CH_4$)-rich gas (50–70%) [14–16], can be upgraded to biomethane (typically > 94% $CH_4$) and used as a fuel for natural gas vehicle (NGVs) or injected into the natural gas grid [17]. China has the largest NGV fleet in the world. The number of NGVs in China has increased from 6000 to 10 million from 2000 to 2020 [18], and the number of light-duty NGVs (LDNGVs) is expected to increase further [19]. Although NGVs are an appropriate method to effectively meet increasingly stringent air pollutant and greenhouse gas emission standards in China [20,21], the environmental effects of LDNGVs operating in China have not been carefully quantified. Previous studies of NGVs using lifecycle assessments (LCAs) [22] have mainly focused on unintended $CH_4$ emissions from the upstream stages (e.g., extraction, processing, and distribution of natural gas) in China [23,24]. By contrast, few studies have been conducted on the LCA of LDNGVs that use FW-derived green methane (GM) over all stages, including the downstream stage, the vehicle manufacturing and operation stages, and the upstream stages.

This paper addresses the following novel challenges that have not been focused in previous researches: (i) developing a large-scale process to produce green $CH_4$ as a LDNGV fuel from FW by using the validated kinetic model based on the effect of FW components on AD, (ii) improving energy efficiency of the whole process including FW derived biogas generation step and green $CH_4$ upgrading step with $CO_2$ adsorption, (iii) performing LCA of the whole process by evaluating process conditions, material and energy balances, and (v) analyzing the environmental impacts as a replacement of disposing FW compared to disposal or energy recovery of FW used in the conventional process. In the present work, a systematic LCA of the new FW scenario (i.e., FW-based GM production and LDNGV operation) in China was performed with consideration of global warming (GW) and fine particulate matter formation (FPMR) environmental impacts. The LCA results for the new FW scenario combined with primary-source-based electricity supply are compared with the LCA results for conventional disposal options of landfilling and incineration. A detailed lifecycle inventory of the FW-based GM production using an AD process was established on the basis of a kinetic study in the present work. This LCA study enables the identification of factors governing the environmental aspects of NGV operation.

## 2. Methods

### 2.1. Process Design: Well-to-Pump

The FW-to-GM process was simulated using the SuperPro Designer software v10 (Figure 1). The FW-to-GM process involves one mixer (MX-101), one reactor (AD-101), two heat exchangers (HX-101 and HX-102), and one separator (GAC-101). Feedstock composition in China was estimated based on the Food and Agriculture Organization (FAO) data, and the processing rate was set to 50 t/d of FW adopting from conventional FW treatment plant. Reaction kinetic equations developed by Angelidaki et al. [25,26] (Table S1), which consider two enzymatic hydrolysis reactions (carbohydrate and protein hydrolysis) and eight bacterial steps (glucose-degrading, lipolytic-degrading, long chain fatty acid-degrading, amino acid-degrading, propionate-degrading, butyrate-degrading, valerate-degrading, and acetoclastic methanogenesis), were used to simulate the AD reactor (AD-101). The feed composition was set to 7 wt% of FW for AD in the AD-101 reactor by mixing FW with $H_2O$ (M-101), which lowered the concentrations of inhibitors during the reaction [25,26]. The feed temperature was set to 328 K for AD conditions (AD-101) by two heat exchangers (HX-101 and HX-102). After the AD process (AD-101), the liquid-phase stream was cooled in HX-101 to preheat the feed stream after MX-101. The minimum approach temperature of heat exchangers was set to 10 K. A bench-scale pressure swing adsorption (PSA) experiment using carbon molecular sieves 3K as the adsorbent was used to simulate the process in the adsorption separator (GAC-101) [27]. The PSA cycle consists of pressurization (70 s), feed (130 s), depress (10 s), and blow (140 s). The crude GM-rich gas-phase stream was then purified to 98.1 wt% GM by removing $CO_2$ and $H_2S$ at the adsorption separator (GAC-101) operating at pressures between 10 and 800 kPa. The

recovery rate of GM from biogas was 79.7%, while the 20.3% of $CH_4$ is lost in the tail gas of the PSA.

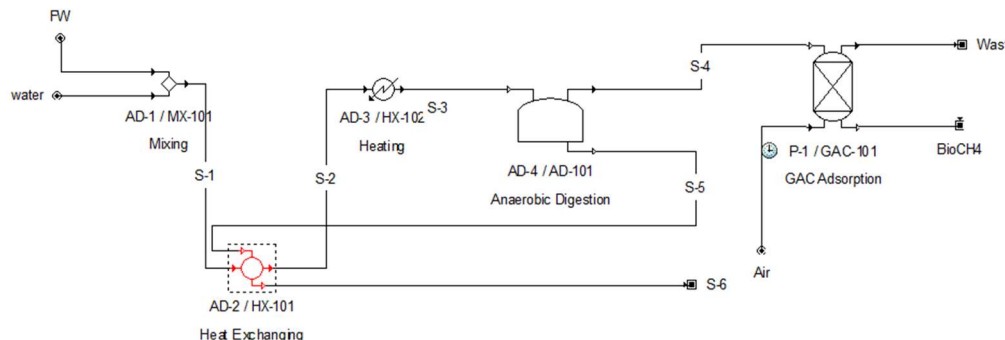

**Figure 1.** Food waste (FW)-to-green methane (GM) process flowsheet created in SuperPro Designer.

*2.2. Life Cycle Assessment*

A LCA of the FW-to-GM process is conducted based on the ISO 14040/14044 standard, which follows four steps: (1) goal and scope definition, (2) inventory analysis, (3) impact assessment, and (4) interpretation [28,29].

The specific goal of the LCA was to determine the environmental impacts of FW-based GM production and LDNGV operation in China. The system boundary for LCA of FW-to-GM process was defined as "well-to-wheel" (Figure 2), combining "FW collection", "well-to-pump" for FW-based GM production, and "pump-to-wheel" for LDNGV operation steps including vehicle manufacturing, operation, and disposal/recycling. A functional unit for the LCA was set to LDNGVs operation using 1.0 kWh of GM.

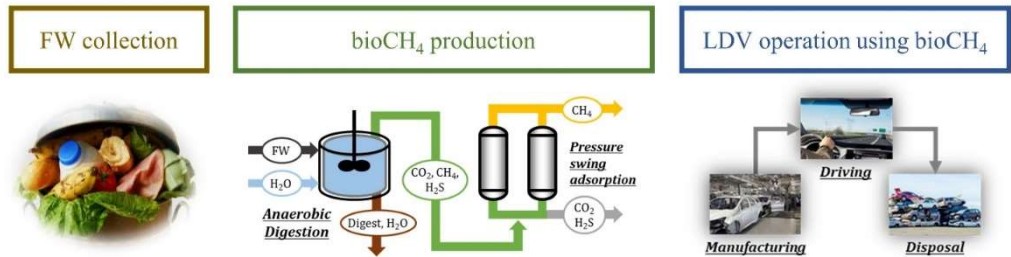

**Figure 2.** System boundaries for the lifecycle assessment of food waste (FW)-based green methane production and light-duty natural gas vehicle (LDNGV) operation in China.

The lifecycle inventory was analyzed based on the materials and energies (M&E) data (i.e., FW, $CO_2$, $CH_4$, heat, and electricity) involved in the process flow diagram (Figure 1). Consequential approach is adapted for the inventory analysis, and the inventory data sources of M&E are the FAO database [4] for FW collection, the Ecoinvent 3 database [30] for well-to-pump, and the greenhouse gases, regulated emissions, and energy in transportation (GREET) database [31] for pump-to-wheel (Table S2).

The environmental impacts of the FW-based GM production and LDNGV operation scenario were evaluated on the basis of the ReCiPe 2016 midpoint-level methodology (Hierarchist perspective) [32] and the GREET model [31]. Previous LCA studies of LDNGV operation using natural gas have focused on analyzing GW for pump-to-wheel on the basis of the potential $CH_4$ emissions [24,33]. In this study, the "well-to-wheel" FW-based GM production and LDNGV operation scenario is compared, with emphasis placed on GW and fine particulate matter formation (FPMR), which are the most common among eighteen impact categories (GW, Stratospheric ozone depletion, Ionizing radiation, Ozone formation, Human health, FPMR, Ozone formation, Terrestrial ecosystems, Terrestrial acidification, Freshwater eutrophication, Marine eutrophication, Terrestrial ecotoxicity Freshwater ecotoxicity, Marine ecotoxicity, Human carcinogenic toxicity, Human non-carcinogenic toxicity, Land use, Mineral resource scarcity, Fossil resource scarcity, Water

consumption) of the ReCiPe 2016 midpoint method [30] and three impact categories (GW, energy use, air pollutants including FPMR) of the GREET model [31].

A sensitivity analysis was conducted for the FW-based GM production and LDNGV operation scenario to identify the effect of primary energy (electricity) sources for well-to-pump on the two GW (GHG-100) and FPMR (PM2.5) impact categories. The primary sources were selected as natural gas (grid mix, base case), wind (alternative case 1), photovoltaic (alternative case 2), and hydroelectric (alternative case 3).

## 3. Results and Discussion

### 3.1. Process Simulation

The M&E input and output of the process for synthesizing GM from FW, as obtained using the SuperPro Designer software, are presented in Table 1. The feedstock consists of 24.4 wt.% of carbohydrate, 7.3 wt.% of protein, 3.2 wt.% of lipid, and 65.1 wt.% of water, which is estimated based on FW generation and nutrient content by food types. The designed process producing 1 kWh GM required 9.7874 kg FW and 0.0932 kWh of heat for its AD reactors and discharged 0.1758 kg of $CO_2$ after the reactors followed by the PSA separators. The electricity requirement is 0.2419 kWh which is required for agitator of AD reactor and compressor of PSA separator. Before energy integration without the heat exchanger (HX-101 in Figure 1), the preheating of the FW stream for the AD reactor (HX-101; 1.6933 kWh of heat) requires more than seven times more energy than the pressurization of the feed stream in the PSA separator. After energy integration with a design that includes a heat exchanger (HX-101 in Figure 2), the total heating requirements decreased by 83.3% (1.4811 kWh of heat). The tail gas of PSA separator contains 20.3% of $CH_4$ produced after AD, and it is combusted for on-site heat supply (0.2038 kWh of heat). Finally, the total heating requirements is decreased to by 94.8% to 0.0932. The results show that preheating the FW stream for the AD reactor requires much less energy (one-third less) than the agitation of feed stream for the AD reactor and the pressurization of the feed stream for the PSA separator. This result implies that electricity is a factor governing the LCA of the FW-based GM production and LDNGV operation scenario. In addition, increase of GM recovery in the PSA separator could be major contributor in the LCA because currently significant amount of $CH_4$ (20% of total $CH_4$ produced from AD) is lost in the tail gas. This have large impact, not only on the environmental impact, but also on the process economics.

**Table 1.** Input–output balance for food waste (FW)-based green methane production in China.

| Input | | | Output | |
|---|---|---|---|---|
| **FW (kg)** | **Heat (kWh)** | **Electricity (kWh)** | **$CO_2$ (kg)** | **$CH_4$ (kWh)** |
| 9.7874 | 0.0932 | 0.2419 | 0.1758 | 0.0649 |

### 3.2. Environmental Effects of the FW-Based GM Production and LDNGV Operation Scenario

The FW-based GM production and LDNGV operation scenario required different amounts of M&E to utilize GM; therefore, we analyzed the influence of these inputs and outputs to determine the major contributors to the GW and FPMR impacts on the scenario (Figure 3). The GW impact for the FW-based GM production and LDNGV operation scenario was 0.98 kg $CO_2$ equiv. per 1 kWh $CH_4$ (Figure 3). Among the three stages, the well-to-pump stage (0.44 kg $CO_2$ equiv.; 44.9% of total GW) was the largest contributor to the GW impact on the scenario, followed by the FW collection (0.31 kg $CO_2$ equiv.; 31.6% of total GW) and the pump-to-wheel (0.23 kg $CO_2$ equiv.; 23.5% of total GW) stages. In the well-to-pump stage, the GW impact was caused mainly by the primary sources required for the electricity generation mix in China (0.25 kg $CO_2$ equiv.; 25.5% of total GW). At present, FW in China is treated using four conventional methods: landfilling (59.4%), incineration (29.5%), dumping (8.1%), and composting (3.0%) [8]. The GW impact of the conventional FW treatment scenario was estimated to be 1.76 kg $CO_2$ equiv. on the basis of the carbon footprint of each conventional FW treatment (as reported by FAO) [4]) and its

usage ratio. Compared with the conventional FW treatment scenario, the new FW-based GM production and LDNGV operation scenario exhibited better environmental impacts, with 44.3% lower GW. As mentioned in the process simulation section, 1 kWh of GM can be produced from 9.7874 kg of FW. When the FW generated in China in 2030 (1.17 billion tons per year) is treated by the FW-based GM production and LDNGV operation scenario, the potential GW reduction would be 93.2 million tons $CO_2$ equiv. per year. According to the China's 2030 Nationally Determined Contribution, the absolute emissions level of China in 2030 is 13.2–13.8 Giga tons $CO_2$ equiv. [34]. The potential GW reduction by introducing new scenario is 0.68–0.71% of China's absolute emissions level in 2030.

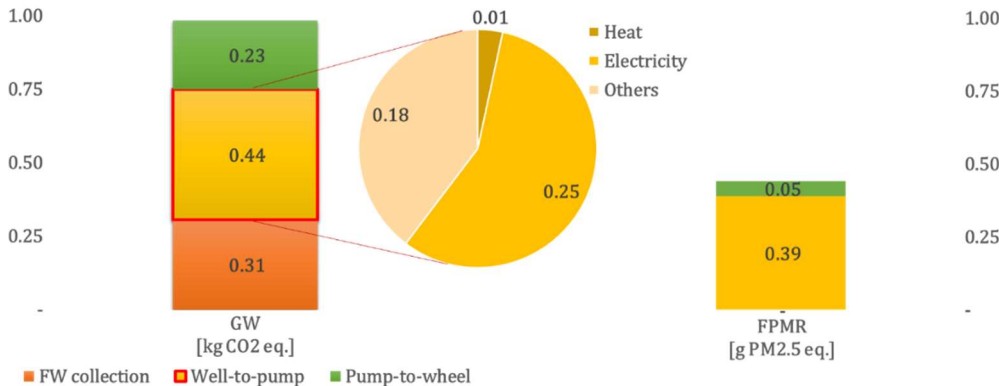

**Figure 3.** Environmental impacts (global warming (GW) and particulate matter formation (FPMR)) of the food waste (FW)-based green methane production and light-duty natural gas vehicle operation scenario (base case).

The FPMR impact, which considered primary emission of fine particulate matter and secondary emission due to $NO_x$ and $SO_2$ emission, for the FW-based GM production and LDNGV operation scenario was 0.44 g PM2.5 equiv. per 1 kWh $CH_4$ (Figure 3). The greatest contributor to the FPMR impact on the scenario was the well-to-pump stage (0.39 g PM2.5 equiv.; 88.6% of total FPMR), followed by the pump-to-wheel stage (0.05 g PM2.5 equiv.; 11.4% of total FPMR). In the well-to-pump stage, the FPMR impact was caused mainly by the primary sources required for the electricity generation mix in China (0.386 g PM2.5 equiv.; 87.7% of total FPMR). The greatest contributor to both the GW and FRS impacts was caused by the primary source required for energy (electricity) use in the well-to-pump stage.

### 3.3. Sensitivity Analysis Varying the Primary Sources

The LCA results for the base case indicates that the environmental impacts (GW and FPMR) of FW-based GM production and LDNGV operation scenario can be strongly influenced by using a different primary source for electricity. Alternative case studies investigating various primary sources (alternative case 1: wind, case 2: photovoltaic, and case 3: hydroelectric) were analyzed in terms of the GM and LDNGV impacts of the new FW scenario (Figure 4). All of the alternative cases had a better effect on GW and FPMR than the base case grid mix: GW (wind (0.74 kg $CO_2$ equiv.); photovoltaic (0.75 kg $CO_2$ equiv.); hydroelectric (0.73 kg $CO_2$ equiv.)); FPMR (wind (0.08 g PM2.5 equiv.); photovoltaic (0.10 g PM2.5 equiv.); hydroelectric (0.06 g PM2.5 equiv.)). Among the alternative cases, the best case is electricity supply from the hydroelectric, while the worst case is electricity supply the photovoltaic. The variations of the FPMR impacts of the new FW scenario were substantially decreased by 78–87% compared with the GW impacts (24–26%). This result implies that the FPMR impacts of the new FW scenario were sensitive to the primary source used for electricity generation.

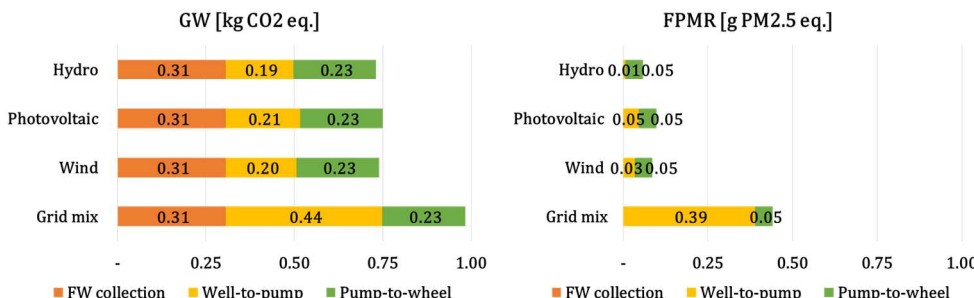

**Figure 4.** Comparison of the environmental impacts (global warming (GW) and particulate matter formation (FPMR)) of different primary sources on energy (electricity) usage.

Although the sensitivity analysis showed that supplying electricity from renewable resources can minimize environmental impacts by indirect emission of well-to-pump stage, the $CO_2$ obtained from AD is still emitted, directly. Additionally, it is hard to reduce the environmental impacts of FW collection requiring fossil energy for FW transportation and pump-to-wheel including direct $CO_2$ emission from $CH_4$ combustion and indirect $CO_2$ emission from vehicle manufacturing and disposal. Unfortunately, it means that the new FW scenario is not a silver bullet for the greenhouse gas reduction. Longer term, we should find novel scenario avoiding direct $CO_2$ emission into atmosphere to achieve carbon neutral.

## 4. Conclusions

An LCA methodology was used to analyze two environmental impacts, GW and FPMR, of a new FW-based GM production and LDNGV operation scenario in China. The LCA results of the new FW scenario combined with electricity supply from a primary source were compared with a conventional FW scenario comprising landfilling, incineration, dumping, and composting. The new FW scenario showed substantially better GW and FPMR impacts than the conventional FW scenario. Supplying electricity to PSA following AD was the key contributor to reducing the environmental impacts of the new FW scenario, and the environmental impacts could be minimized by supplying electricity from renewable resources. The results showed that the new FW scenario could be a feasible option for reducing GW and FPMR in the short term. Longer term, a novel approach without the concern of unavoidable $CO_2$ emission is required.

**Supplementary Materials:** The following supporting information can be downloaded at: https://www.mdpi.com/article/10.3390/fermentation8120680/s1, Table S1: Stoichiometry and kinetic equations of the anaerobic digestion for bio$CH_4$ production; Table S2: Details of life cycle inventory database for FW-based GM production in China.

**Author Contributions:** Conceptualization, J.H.; methodology, J.H. and J.B.; investigation, J.B.; writing—original draft preparation, J.B. and J.H.; writing—review and editing, J.B. and J.H. All authors have read and agreed to the published version of the manuscript.

**Funding:** This work was supported by a National Research Foundation of Korea (NRF) grant funded by the Korea Government (MSIT; No. 2022R1C1C1003329) and the program of Development of Eco-friendly Chemicals as Alternative Raw Materials to Oil through the National Research Foundation of Korea (NRF) funded by the Ministry of Science and ICT (2022M3J5A1085257).

**Institutional Review Board Statement:** Not applicable.

**Informed Consent Statement:** Not applicable.

**Conflicts of Interest:** The authors declare no conflict of interest.

## Nomenclatures

| | |
|---|---|
| AD | Anaerobic digestion |
| FPMR | Fine particulate matter formation |
| FW | Food waste |
| GM | FW-derived green methane |
| GHG | Greenhouse gas |
| GREET | The greenhouse gases, regulated emissions, and energy in transportation |
| LCAs | Lifecycle assessments |
| LDNGVs | Light-duty NGVs |
| M&E | Materials and energies |
| CH4 | Methane |
| NGVs | Natural gas vehicle |
| PSA | Pressure swing adsorption |

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
