# Peer review of "Green Methane as a Future Fuel for Light-Duty Vehicles"

_fermentation, doi:10.3390/fermentation8120680_

Round 1

Reviewer 1 Report

The manuscript converts food waste into green methane through anaerobic digestion and conducts a life cycle assessment of food waste-based green methane production and light-duty natural gas vehicles operation in China. In general, the manuscript can give certain help in reducing the environmental impact of the food waste treatment process. However, there are still some issues.

1.      In Section 3.1, how is 83.3% calculated?

2.      Some abbreviations of professional terms appear in the text, and their meanings should be annotated to improve the readability of the article.

3.      The fine particulate matter formation impact of wind is 0.08 g PM2.5 equiv. in Figure 4, which is inconsistent with the description in section 3.3.

Author Response

We appreciate the reviewer’s comment. We carefully revised our manuscript by considering reviewer’s comments. Please find the attached file.

Reviewer 2 Report

Dear authors, the study is of interest, but the LCA is not totally correct

·          The FU is not well designed and consistent with the study

·       The scope of the study is not provided

·       You should provide a detailed inventory for the phase of LCI

·       Please explain if you adopted allocation or system expansion

·       You should define what is included or not in foreground and background

Some study that can help you are following reported:

1.     The application of Life Cycle Assessment to Integrated Solid Waste Management. Part 1 - Clift, R., Doig, A., Finnveden, G., 2000. Methodology. Process Saf. Environ. Prot. 78, 279–287. https://doi.org/10.1205/095758200530790

2.     Life cycle assessment and life cycle costing of advanced anaerobic digestion of organic fraction municipal solid waste F. Demichelis, T. Tommasi, F.A. Deorsola, D. Marchisio, G. Mancini, D. Fino PII: S0045-6535(21)03530-X DOI: https://doi.org/10.1016/j.chemosphere.2021.133058

Author Response

(The authors gave the same response as above.)

Round 2

Reviewer 2 Report

Tha manuscript is really improved and well done.